# Adaptive substitutions underlying cardiac glycoside insensitivity in insects exhibit epistasis in vivo

Andrew M Taverner[1], Lu Yang[2], Zachary J Barile[3,4], Becky Lin[3,4], Julie Peng[1], Ana P Pinharanda[5], Arya S Rao[5], Bartholomew P Roland[3,4], Aaron D Talsma[3,4], Daniel Wei[3,4], Georg Petschenka[6], Michael J Palladino[3,4]*, Peter Andolfatto[5]*

[1]Lewis-Sigler Institute for Integrative Genomics, Princeton University, Princeton, United States; [2]Department of Ecology and Evolutionary Biology, Princeton University, Princeton, United States; [3]Department of Pharmacology and Chemical Biology, University of Pittsburgh, Pittsburgh, United States; [4]Pittsburgh Institute for Neurodegenerative Diseases (PIND), University of Pittsburgh School of Medicine, Pittsburgh, United States; [5]Department of Biological Sciences, Columbia University, New York, United States; [6]Institute for Insect Biotechnology, Justus-Liebig-Universität Gießen, Hesse, Germany

**Abstract** Predicting how species will respond to selection pressures requires understanding the factors that constrain their evolution. We use genome engineering of *Drosophila* to investigate constraints on the repeated evolution of unrelated herbivorous insects to toxic cardiac glycosides, which primarily occurs via a small subset of possible functionally-relevant substitutions to $Na^+,K^+$-ATPase. Surprisingly, we find that frequently observed adaptive substitutions at two sites, 111 and 122, are lethal when homozygous and adult heterozygotes exhibit dominant neural dysfunction. We identify a phylogenetically correlated substitution, A119S, that partially ameliorates the deleterious effects of substitutions at 111 and 122. Despite contributing little to cardiac glycoside-insensitivity in vitro, A119S, like substitutions at 111 and 122, substantially increases adult survivorship upon cardiac glycoside exposure. Our results demonstrate the importance of epistasis in constraining adaptive paths. Moreover, by revealing distinct effects of substitutions in vitro and in vivo, our results underscore the importance of evaluating the fitness of adaptive substitutions and their interactions in whole organisms.

DOI: https://doi.org/10.7554/eLife.48224.001

*For correspondence:
mjp44@pitt.edu (MJP);
pa2543@columbia.edu (PA)

**Competing interests:** The authors declare that no competing interests exist.

## Introduction

Understanding the factors that limit the rate of adaptation is central to our ability to forecast future adaptive evolutionary trajectories and predict the timescales over which these changes are expected to occur (*Stern, 2011*; *Losos, 2017*; *Morris et al., 2018*). In particular, considerable uncertainty surrounds the relative importance of the availability of adaptive mutations, pleiotropy and epistasis in constraining adaptive paths (*Stern, 2011*; *Storz, 2018*). One fruitful approach to addressing this question has been to examine repeated bouts of adaptation in microbial systems subject to a common selective pressure and identical starting conditions (*Jerison and Desai, 2015*). Unfortunately, such approaches still have limited utility in multicellular eukaryotes and likely do not reveal the full range of constraints operating in nature. An alternative and analogous approach is to examine evolutionary patterns in large, naturally occurring assemblages of species exhibiting parallel adaptations in response to a common selective pressure (*Liu et al., 2010*; *Meyer et al., 2018*; *Zhen et al., 2012*; *Dobler et al., 2012*; *Christin et al., 2007*). Evolutionary studies of parallelisms are a powerful

complementary approach to deducing the factors constraining adaptation (*Stern, 2013*). A well-known example of parallel adaptations is the ability of numerous animals to acquire toxins from their environments and sequester them for use in defense against predators (*Brodie, 2009*; *Erb and Robert, 2016*).

Here, we focus on a large group of herbivorous insects with a broad phylogenetic distribution that have independently specialized on toxic host plants (*Dobler et al., 2011*). In addition to other defenses against herbivory, the Apocynaceae and other plant species produce a class of toxic secondary compounds called cardiac glycosides (CGs). CGs are highly toxic to animals because they are potent inhibitors of $Na^+,K^+$-ATPase (NKA), a ubiquitously expressed enzyme needed in a variety of cellular processes in animals, including neural signal transduction, muscle contraction, and osmoregulation (*Lingrel, 2010*). Mutations to NKA in invertebrates are typically homozygous lethal and associated with defects in locomotion, neuron development and neural homeostasis (*Ashmore et al., 2009*). In humans, loss-of-function mutations in NKA have been associated with several rare disorders such as dystonia, parkinsonism and hemiplegic migraines (*Bøttger et al., 2012*). Despite their toxicity, NKAs have long been targeted with CG-based drugs to treat common conditions such as congestive heart failure and cardiac arrhythmias (*Schoner, 2002*).

Insensitivity to CGs in insects can evolve via several mechanisms including modification of the CG-binding domain of NKA (i.e. target-site insensitivity), restriction of NKA expression to neurons (*Petschenka et al., 2013b*), the deployment of proteins that ameliorate the toxic effects of CGs (*Torrie et al., 2004*; *Petschenka et al., 2013b*) and other physiological factors (*Vaughan and Jungreis, 1977*). Despite this wide variety of potential paths to CG-insensitivity, the evolution of insensitivity in most CG-adapted insects is due, at least in part, to target-site insensitivity. Indeed, in most cases, diet specialization on CG-containing hostplants has been accompanied by recurrent adaptive amino acid substitutions to the CG-binding domain of the alpha-subunit of NKA, ATPα1 (*Zhen et al., 2012*; *Dobler et al., 2012*; *Yang et al., 2019*). Previous studies have identified up to 35 sites in ATPα1 at which substitutions could contribute to CG-insensitivity (reviewed in *Zhen et al., 2012*). However, CG-insensitivity of ATPα1 most often arises via a highly similar pattern of substitution at two sites (111 and 122, *Figure 1A*): as an illustration, a survey of 28 CG-adapted insects revealed that 30 of 63 amino acid substitutions observed at sites implicated in CG-sensitivity in ATPα1 occur at sites 111 and 122 (*Yang et al., 2019*). Sites 111 and 122 have also been identified as targets of positive selection in CG-adapted insects using statistical phylogenetic methods (*Yang et al., 2019*). Understanding why these two sites, in particular, are so often employed requires a characterization of the effects of these substitutions, individually and in combination, on organismal phenotypes and fitness.

To explain the frequent reuse of sites 111 and 122, it has been speculated that substitutions at most alternative sites may be associated with negative pleiotropic effects, that is, have deleterious effects on another aspect of phenotype and fitness (*Zhen et al., 2012*). Support for this hypothesis comes from the fact that multiple insect species specializing on CG-containing host-plants have independently duplicated and neofunctionalized ATPα1. In all cases examined to date, species with two or more copies retain one minimally altered copy that is more highly expressed in nervous tissue, and have evolved one or more insensitive copies that are more highly expressed in the gut, the site of absorption of CGs (*Zhen et al., 2012*; *Yang et al., 2019*). Further support for negative pleiotropic effects is provided by the expression of engineered ATPα1 constructs in cell lines, suggesting that some duplicate-specific CG-insensitivity substitutions appear to reduce NKA activity (*Dalla and Dobler, 2016*). Based on these findings, the frequent parallel substitutions observed at sites 111 and 122 in specialists lacking duplicate ATPα1 copies plausibly reflect the fact that substitutions at these sites are minimally pleiotropic.

## Results

### Common substitutions at positions 111 and 122 exhibit negative pleiotropic effects

To test the idea that substitutions at positions 111 and 122 lack strong negative pleiotropic effects, we used the transgenesis toolkit of *Drosophila melanogaster* (*Figure 1—figure supplement 1*), a generalist insect that harbors a single ubiquitously expressed copy of a CG-sensitive form of ATPα1.

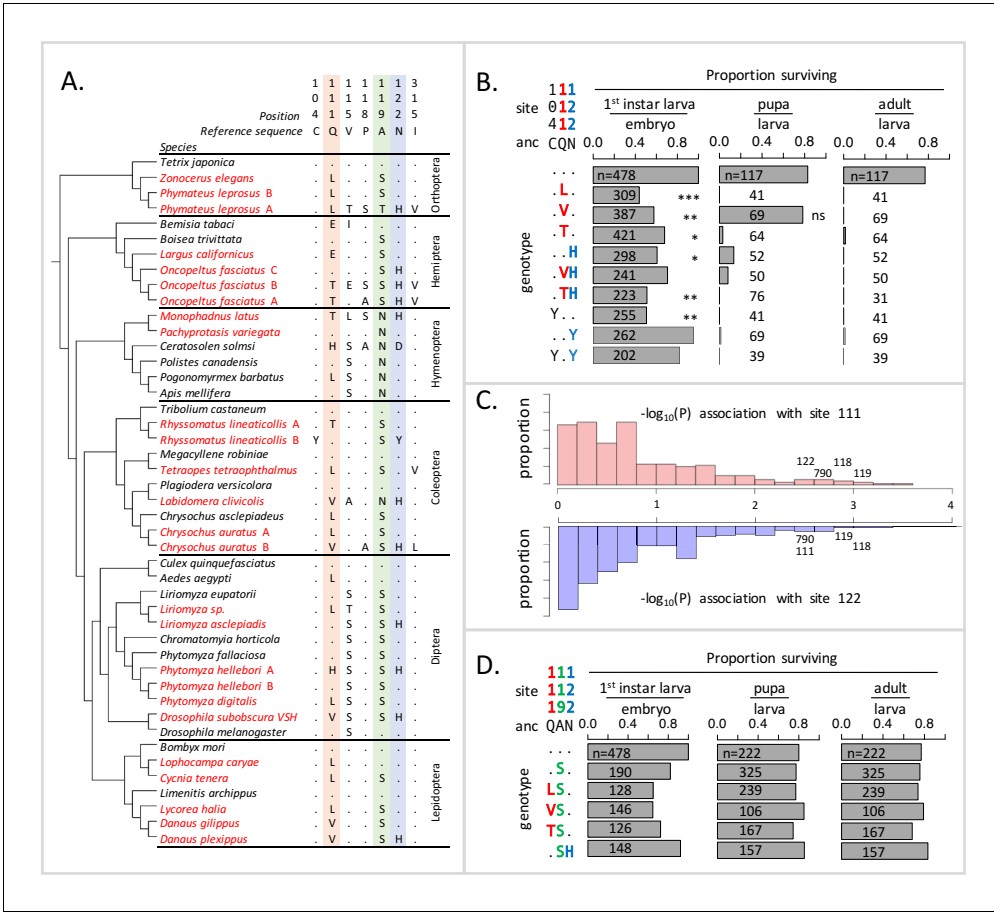

**Figure 1.** Deleterious effects of substitutions at positions 111 and 122 are ameliorated by the permissive substitution A119S. (**A**) Patterns of amino acid substitution at sites implicated in CG sensitivity near the H1-H2 transmembrane domain of ATPα1 in representative species from six insect orders (see *Supplementary file 1* for all species). CG-adapted species names are shown in red. Dots indicate identity with the ancestral reference sequence and letters indicate derived amino acid substitutions. Positions 111 and 122, highlighted in pink and blue respectively, are hotspots of frequent parallel substitution. C104Y and N122Y represent rare substitutions associated with ATPα1 duplication (*Zhen et al., 2012*). Position 119 (highlighted in green) had not been previously implicated in CG sensitivity, but is identified as a candidate permissive substitution in subsequent analyses. (**B**) Viability of homozygous first instar larvae, pupae and adults for the first series of engineered substitution lines. Dots indicate identity with the ancestral reference at sites C104, Q111, and N122. Plotted is the proportion of surviving homozygous mutant offspring (i.e. EYFP-), scaled by the proportion for the wild-type control. The number of individuals assayed is indicated in each case. Survival odds relative to the wild type-strain (. . .) was tested using a Fisher's Exact Test, with adjusted P-values indicated as: \*\*\*p<0.001; \*\*p<0.01, \*p<0.05. All larvae to pupa survivorship are significantly lower than wild-type (p<1e-5) except for Q111V. All larvae to adult survivorship are significantly lower than wild-type (p<1e-5). (**C**) Distributions of P-values for the strength of the phylogenetic correlation between sites 111 (above) or 122 (below) and 270 non-singleton amino acid variants in a multi-sequence alignment including 174 ATPα1 sequences representing 161 species. Sites 111 and 122 exhibit highly correlated evolution. Three additional sites in the lowest 5% of P-values shared in common between sites 111 and 122, including site 119, are labeled. (**D**) Viability of homozygous first instar larvae, pupae and adults for the second series of engineered substitution lines that include A119S relative to the wild-type control. Dots indicate identity with the ancestral reference at sites Q111, A119, and N122. The number of individuals assayed is indicated in each case. Despite slight reductions in viability, all reductions are not significant relative to wild-type line after multiple test correction.

DOI: https://doi.org/10.7554/eLife.48224.002

The following figure supplements are available for figure 1:

**Figure supplement 1.** Engineering strategy for generating amino acid substitution lines.

DOI: https://doi.org/10.7554/eLife.48224.003

*Figure 1 continued*

**Figure supplement 2.** Phylogeny showing relationships of the sampled species.
DOI: https://doi.org/10.7554/eLife.48224.005
**Figure supplement 3.** Variant sites in ATPα1 most strongly correlated with substitutions with site 119.
DOI: https://doi.org/10.7554/eLife.48224.006
**Figure supplement 4.** Crystal structure of NKA bound to the cardiac glycoside ouabain (PDB:4HYT).
DOI: https://doi.org/10.7554/eLife.48224.007

We focus on several substitutions at sites 111 and 122 (notably Q111V, Q111T, N122H, *Figure 1A*) that have been directly implicated in CG-insensitivity in functional experiments (reviewed in *Zhen et al., 2012*). By engineering amino acid substitutions into a single *D. melanogaster* background, we ensure that fitness differences observed among lines are caused by the substitution and not confounded by unknown variation in the genomic background. In addition, we can rule out compensatory changes elsewhere in the genome, or evolved changes in physiology, which are concerned in multi-generation population-level experimental evolution studies. Importantly, by testing substitutions in vivo, we can evaluate their functional effects at multiple phenotypic levels, from the biochemistry of enzyme inhibition to behavior and fitness. We generated six lines that carry substitutions at these two sites of the endogenous ATPα1 locus individually (Q111L, Q111V, Q111T, N122H) and in combination (Q111V+N122H, Q111T+N122H) (see Materials and methods). For comparison, we also created three lines in which we introduced two rare, copy-specific substitutions (C104Y and N122Y and C104Y+N122Y, *Figure 1A*) for which we did not have a priori expectations about pleiotropic fitness effects. C104Y is known to confer some degree of insensitivity to CGs, and both C104Y and N122Y occur on a neofunctionalized copy of ATPα1 in the milkweed weevil (*Zhen et al., 2012*). An additional control line carrying the wild-type *D. melanogaster* allele was generated using the same approach.

Based on the repeated use of substitutions at positions 111 and 122, we expected these would confer ATPα1 with some degree of insensitivity to CGs and be associated with either no or mild negative pleiotropic effects on fitness. To our surprise, however, each of the nine lines exhibit severely reduced fitness, behaving effectively as recessive lethals. To identify the developmental stage at which lethality occurs, we evaluated hatchability (i.e. the proportion of homozygous larva per embryo) and the probability of individuals surviving to pupae and adults (*Figure 1B*). Severe fitness deficits for most engineered substitution lines were apparent as early as the larval hatching stage. Still, for most engineered mutant lines, greater than 50% of individuals hatch into the first instar. Q111V homozygotes exhibit particularly high probabilities of survival until the pupal stage. Nonetheless, survivorship of individuals to adulthood is close to zero for all lines.

## The substitution A119S rescues lethality of substitutions at sites 111 and 122

The unexpected deleterious effects of common substitutions at sites 111 and 122 raises the question of how insects with sensitive ATPα1 isoforms, spanning a broad phylogenetic distribution, can evolve insensitivity to CGs via substitutions to one or both of these sites. An important clue is provided by the observation that *D. subobscura*, a European relative of *D. melanogaster,* carries haplotypes resembling those that we engineered into *D. melanogaster*, including the ancestral (QN) and derived states (QH and VH) at sites 111 and 122, respectively (*Pegueroles et al., 2016*), and yet are viable. PAML ancestral sequence reconstruction applied to full length coding sequences for *D. subobscura* (Materials and methods) and the closely related sister species *D. guanche* (*Puerma et al., 2018*) reveals that 21 amino acid substitutions distinguish the *D. melanogaster* ATPα1 from the ancestral *D. subobscura* protein. We hypothesized that one or more of these substitutions ameliorates the deleterious effects of substitutions at positions 111 and 122.

Using a multiple sequence alignment that includes 161 CG-adapted insects and outgroup species surveyed from six insect orders (*Figure 1—figure supplement 2* and *Supplementary file 1*, *Figure 1A* shows a subset of these taxa), we applied the software BayesTraits (*Pagel and Meade, 2006*) to evaluate the evidence for correlated evolution between sites 111 and 122 and all other variant sites in the protein (Materials and methods). Of the divergent sites between *D. melanogaster* and *D. subobscura*, only one site (119) is among those having the strongest phylogenetic

correlations with both sites 111 and 122 (top 5% of 270 sites) (*Figure 1C*, *Figure 1—figure supplement 3*). A substitution at 119 is observed in 90% of the cases in which there is a substitution at 111 and 100% of the cases in which there is a substitution at site 122. *D. melanogaster* retains the ancestral Alanine at this site, whereas the *D. subobscura* and closely-related species harbor a derived Serine substitution.

Though site 119 is not among sites known to affect CG-sensitivity (*Zhen et al., 2012*), previous work identified it as one of four sites in the protein that underlie a CG-association/dissociation rate difference distinguishing human ATP1A1 and ATP1A2 isoforms (*Crambert et al., 2004*). Considering the orientation of the amino-acid side chain of site 119 with respect to a bound CG (*Figure 1—figure supplement 4*), it is unlikely that it plays a direct role in CG binding. Nonetheless, due to its physical proximity to sites 111 and 122 and the evidence for correlated evolution with these sites, we hypothesized that the A119S substitution may compensate for the deleterious effects of substitutions at positions 111 and 122. To test this hypothesis, we generated a second series of substitution lines that include A119S in isolation, and A119S paired with four substitutions at sites 111 and 122 (i.e. Q111L+A119S, Q111V+A119S, Q111T+A119S and A119S+N122H, *Figure 1D*). Embryos homozygous for A119S, with and without substitutions at 111 and 122, have levels of hatchability and survival that are close to wild-type levels. Remarkably, we find that A119S rescues the lethality associated with homozygosity for all four substitutions with which we paired it (i.e. compare *Figure 1D* with *Figure 1B*). These results establish the existence of epistatic fitness interactions between A119S and multiple substitutions at both 111 and 122.

## A119S rescues enzyme dysfunction associated with substitutions at sites 111 and 122

To gain insight into the functional basis of the fitness interaction between A119S and substitutions at sites 111 and 122 at the level of NKA function, we carried out a series of enzyme inhibition assays (Materials and methods, *Figure 2*). Ideally, we would like to directly compare activities and inhibition profiles for individual substitutions at positions at 111 and 122 alone and in combination with A119S in vivo by fitting monophasic enzyme-inhibition curves (e.g. *Petschenka et al., 2013a*). However, homozygous lethality associated with substitutions at positions 111 and 122 in the absence of A119S prevents us from using this approach. Instead, we opted to compare wild-type and mutant forms of NKA in heterozygous (i.e. Mut/+) individuals, where we expect a heterogeneous mixture of two isoforms. In this case, assuming the mutant form of the enzyme is reasonably active (i.e. $f > 0$), we expect to see biphasic inhibition curves reflecting the equilibrium dissociation constants ($K_d$) corresponding to the wild-type (+) and mutant (Mut) forms of the enzyme, respectively (see Materials and methods and *Figure 2—figure supplement 1*). If the mutant form of the enzyme is inactive, $f = 0$ and would result in a monophasic curve (Materials and methods, *Equation 1*). Alternatively, failure to detect a biphasic curve may also suggest that the $K_d$ of the wild-type and mutant forms of the enzyme are too close to be distinguished.

With these considerations in mind, we note that the inhibition curve for A119S/+ appears to be monophasic (*Figure 2A*). Since A119S homozygotes are viable, we conclude that A119S alone likely encodes a functional enzyme that has little effect on CG inhibition of NKA. In contrast, the inhibition curve for N122H/+ heterozygotes is biphasic ($f = 0.18$, 95% CI 0.14–0.24). However, while this implies that N122H substantially increases CG-insensitivity ($IC_{50,2} = 6.6e-6$), the contribution of the N122H form to the total CG-inhibitable activity in heterozygotes ($f$) is estimated to be less than half that of the wild-type form. An analysis of allele-specific expression of N122H/+ indicates close to equal mRNA levels for the two alleles (*Figure 2—figure supplement 2*), suggesting that the differences in activity are not due to differences at the level of gene expression. Additionally, unlike A119S, N122H alone results in homozygous lethality. Thus, we conclude that, despite conferring CG-insensitivity, the N122H substitution likely encodes a functionally-impaired enzyme.

In contrast to N122H/+, N122H+A119S/+ heterozygotes produce a strongly biphasic inhibition curve, with a two-fold higher estimated $IC_{50,2} = 1.9e-5$ and comparable levels of activity to that of the wild-type form ($f = 0.53$, 95% CI 0.47–0.56, *Figure 2A*). Similar results were obtained in comparisons of Q111V and Q111V+A119S (*Figure 2—figure supplement 3*). We estimate the level of CG-insensitivity conferred by Q111L, Q111V, Q111T and N122H, in the presence of A119S, to be 7, 11, 28 and 178-fold greater than the wild-type form of the enzyme, respectively (*Figure 2B*). Our results demonstrate epistasis between substitutions at sites 119 and 111/122 at the level of enzyme

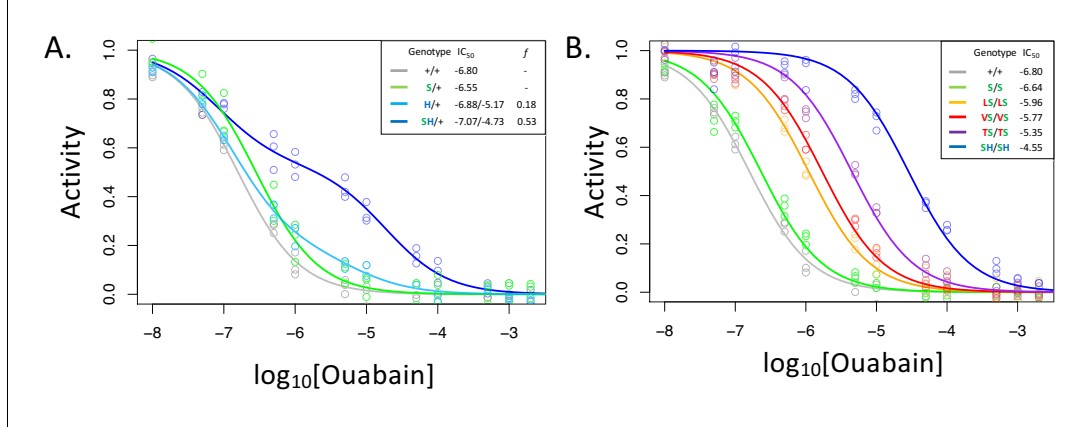

**Figure 2.** A119S ameliorates detrimental effects of CG-insensitivity substitutions on NKA function. Plotted are relative activity as a function of increasing concentrations of the cardiac glycoside (CG) ouabain. (**A**) Inhibition curves are plotted for heterozygous individuals to allow comparison of the effects of these substitutions in the presence and absence of A119S. Points represent biological replicates, each of which is the mean across three technical replicates. Curves for the engineered wild-type strain (+/+) and A119S/+ (S/+) are plotted for comparison. The presence of A119S alone confers a negligible increase in CG-insensitivity and results in a monophasic curve. In contrast, N122H/+ (H/+) and A119S+N122H (SH/+) exhibit biphasic curves (i.e. the estimated proportion of CG-inhibitable activity of the mutant form, $f$, are significantly larger than 0). For H/+, $f$ = 0.18 (95% CI 0.14–0.24), and is significantly lower than that of SH/+, $f$ = 0.53 (95% CI 0.47–0.56). (**B**) Inhibition curves for homozygous substitutions Q111L, Q111V, Q111T and N122H in the presence of A119S reveal that they increase CG-insensitivity by 7, 11, 28 and 178-fold, respectively. The effect of A119S alone is estimated to be 1.45-fold relative to +/+ (95% CI 1.13–1.86).

DOI: https://doi.org/10.7554/eLife.48224.008

The following figure supplements are available for figure 2:

**Figure supplement 1.** Biphasic inhibition curve fitting allows the estimation of relative activities of two enzyme forms in a mixture.
DOI: https://doi.org/10.7554/eLife.48224.009

**Figure supplement 2.** Allele-specific expression (ASE) in heterozygous lines.
DOI: https://doi.org/10.7554/eLife.48224.010

**Figure supplement 3.** Enzyme inhibition curves for Q111V (V) with and without A119S (S).
DOI: https://doi.org/10.7554/eLife.48224.011

function. Previous studies have demonstrated increased survival of cell lines expressing Q111V, N122H, and Q111T+Q111H (*Dobler et al., 2012*), and substantial insensitivity to CG-inhibition despite little effect on ATPase activity for N122H, Q111T+Q111H and Q111V+Q111H (*Dalla et al., 2013*; *Dalla and Dobler, 2016*). Given that these substitutions were engineered on a *D. melanogaster* background lacking A119S, our results suggest such substitutions are likely to be associated with substantial enzyme dysfunction.

## A119S rescues neural dysfunction associated with substitutions at sites 111 and 112 in vivo

An advantage of functionally testing the effects of amino acid substitutions in vivo, as opposed to in vitro, is the ability to examine fitness-related phenotypes at multiple levels. We therefore considered the impact of substitutions, alone and in combination, on higher level fitness-related phenotypes including adult behavior in response to stress and survivorship of adult flies exposed to CGs.

NKA plays a central role in maintaining neuron action potentials and previous studies have documented short-term paralysis following mechanical overstimulation (aka 'bang sensitivity') associated with mutations that reduce this enzyme's activity (*Ganetzky and Wu, 1982*; *Schubiger et al., 1994*; *Davis et al., 1995*; *Palladino et al., 2003*). As such, we used the bang sensitivity phenotype as a proxy for proper neural function. Examining our panel of substitutions for these effects reveals all of the substitutions we engineered at sites 111 and 122, individually and in combination, exhibit dominant bang sensitivity phenotypes that are similar in severity to those of the loss-of-function deletion Δ2-6b (*Figure 3A*, *Figure 3—figure supplement 1*). In contrast, bang sensitivity phenotypes were indistinguishable from wild-type for all heterozygous substitutions at 111 and 122 in combination with A119S (*Figure 3A*, *Figure 3—figure supplement 2*). Interestingly, individuals homozygous for

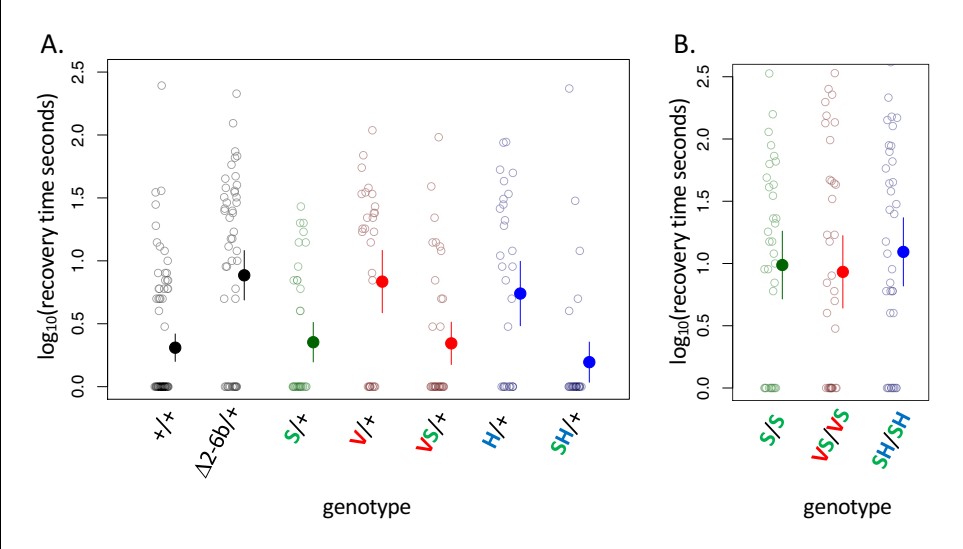

**Figure 3.** A119S ameliorates detrimental effects of CG-insensitivity substitutions on neural function. Plotted are recovery times for individuals (open circles), and means with approximate 95% confidence bounds (solid circles with whiskers), following mechanical over-stimulation (aka, the 'bang sensitivity' assay). Flies heterozygous for loss-of-function ATPα1 substitutions are known to have significantly longer recovery times than wild-type flies (see for e.g. Δ2-6b/+ relative to +/+ in panel A, p=4.5e-7). (A) Distributions of recovery times for heterozygous flies (i.e. Mut/+) carrying individual substitutions A119S (S/+), Q111V (V/+) and N122H (H/+) and combinations VS/+ and SH/+. The recovery-time distributions for S/+, VS/+ and SH/+ are indistinguishable from the engineered wild-type strain (+/+). In contrast, recovery times are significantly longer than wild-type for lines with individual substitutions V/+ and H/+ (p=1.3e-5 and p=5.2e-4, respectively) and are indistinguishable from the loss-of-function deletion mutation Δ2-6b/+. (B) Recovery times for homozygous lines S/S, VS/VS and SH/SH are all significantly longer than for heterozygotes S/+, VS/+, and SH/+ (p<3.3e-3). These results demonstrate epistasis between substitutions at sites 119 and 111/122 at the level of neural function. A119 ameliorates the dominant dysfunction caused by Q111V and N122H but residual recessive neural dysfunction is still apparent in homozygotes that include A119S.
DOI: https://doi.org/10.7554/eLife.48224.012

The following figure supplements are available for figure 3:

**Figure supplement 1.** Bang sensitivity phenotypes of lines with heterozygous substitutions at sites 104, 111 and 122.
DOI: https://doi.org/10.7554/eLife.48224.013

**Figure supplement 2.** Bang sensitivity phenotypes of substitutions at sites 111 and 122 on the background of A119S.
DOI: https://doi.org/10.7554/eLife.48224.014

A119S, and substitutions at 111 and 122 in the presence of A119S, still exhibit obvious neural dysfunction relative to the wildtype (*Figure 3*, *Figure 3—figure supplement 2*). Thus, while A119S is itself associated with recessive pleiotropic effects, our results demonstrate some degree of positive epistasis between A119S and substitutions at sites 111 and 122 when considering the level of adult behavior.

## Substitutions at sites 111, 119 and 122 increase adult survival upon exposure to CGs

*D. melanogaster* do not normally consume CG-containing plants and consumption of CGs results in increased mortality (*Groen et al., 2017*). Given that substitutions at sites 111 and 122 decrease sensitivity to CG-inhibition of NKA, such substitutions should confer an advantage upon exposure of *D. melanogaster* to CGs. To test this hypothesis, we exposed adult animals to media containing the CG ouabain. While the wild-type strain suffers high levels of mortality upon CG exposure, the lines carrying substitutions at sites 111 and 122 in combination with A119S are all substantially less-sensitive (*Figure 4*, *Figure 4—figure supplement 1*). Notably, the survival probability of A119S+N122H (SH/+ and SH/SH) is indistinguishable from control treatments in exposures with up to 10 mM ouabain.

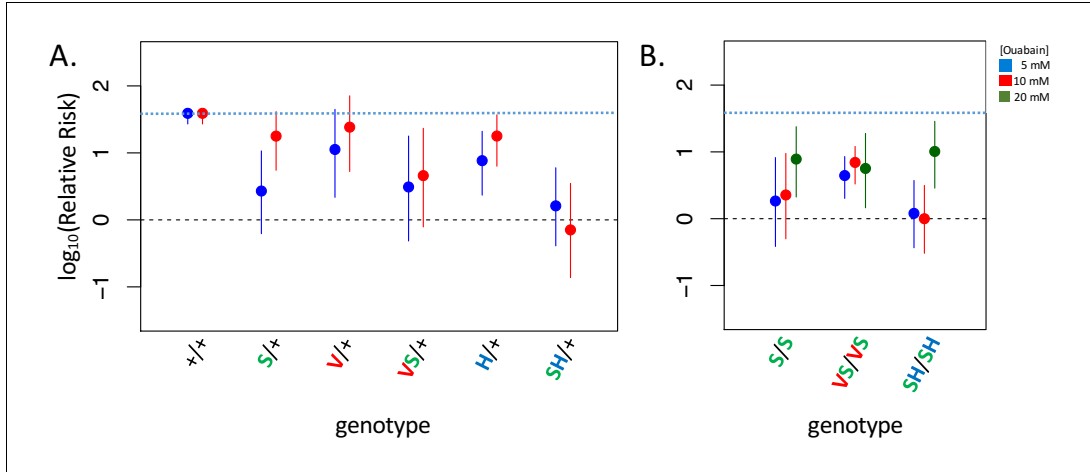

**Figure 4.** Adult survival upon 7 day exposure to the CG ouabain. Plotted is the log relative risk for treatments (5, 10 or 20 mM ouabain) relative to no treatment controls (no ouabain) for (**A**) heterozygous strains and (**B**) homozygous strains. Estimates (points) and 95% confidence bounds (whiskers) were obtained using the Cochran-Mantel-Haenzel framework. Each estimate is based on three biological replicates of 20 flies per concentration. A value of 0 corresponds to equal probability of survival on treatment versus the no ouabain control indicating complete insensitivity to a given concentration of ouabain. Labels: + = engineered wild-type allele; S = A119S; V = Q111V; VS = Q111V+A119S; H = N122H; SH = A119S+N122H. These results reveal appreciable CG-insensitivity conferred by individual substitutions A119S (S/+ and S/S) and the recessive lethals Q111V (V/+) and N122H (H/+) compared to the engineered wild-type control strain (+/+).
DOI: https://doi.org/10.7554/eLife.48224.015

The following figure supplement is available for figure 4:

**Figure supplement 1.** Adult survival of all homozygous strains with A119S upon 7 day exposure to CGs.
DOI: https://doi.org/10.7554/eLife.48224.016

Beyond confirming an association between insensitivity to CG-inhibition of enzyme activity in vitro and reduced sensitivity to CG-exposure in vivo, two additional important findings arise from these experiments. First, a substantial measure of insensitivity to CG-exposure is also conferred by A119S alone (S/+ and S/S exhibit ~8.5 fold and ~21 fold lower relative risk, respectively, than wild-type at 5 mM ouabain, *Figure 4*). Despite this, A119S alone has only a small effect on NKA sensitivity to CG-inhibition (*Figure 2*). Further, the substantially improved survival of heterozygous strains relative to the wild-type strain, including to some extent A119S/+, suggests that insensitivity to CG exposure is a partially dominant phenotype (*Figure 4*, *Figure 4—figure supplement 1*). Importantly, despite the lethal homozygous effects of Q111V and N122H individually (*Figure 1B*), the dominant effects of these substitutions on insensitivity to CG-exposure imply that they have the potential to confer a fitness advantage as heterozygotes in CG-rich environments.

## Discussion

Our findings carry important implications for how CG-insensitivity evolves in insects. At the onset of this study, we expected minimal negative pleiotropic effects associated with adaptive substitutions at sites 111 and 122, given their frequent parallel occurrence across insect orders (*Zhen et al., 2012*). Our findings that these substitutions are associated with negative pleiotropic effects at multiple levels (i.e. enzyme function, neural function and viability) in *D. melanogaster* suggests a potentially more complicated explanation. As we show, with few exceptions, adaptive substitutions conferring CG-insensitivity at sites 111 and 122 most often arise on a genetic background that includes a substitution at site 119, a site not previously implicated in CG-sensitivity. We further show that the most commonly observed substitution at site 119, A119S, rescues the lethality and dominant effects on neural function associated with substitutions at sites 111 and 122 in *D. melanogaster*. A second common substitution, A119N, appears independently in three insect orders and may act similarly (*Figure 1A*, *Supplementary file 1*). Thus, the repeated use of specific substitutions does

not mean that these changes unconditionally lack negative pleiotropic consequences and that their fitness advantage can depend critically on the background on which they arise.

It is notable that many of the amino acid substitutions associated with CG-adaptation (i.e. Q111L, Q111E, Q111H, A119S, and N122H), can all be reached by single nucleotide substitutions, whereas others (e.g. Q111V, Q111T) require two nucleotide substitutions. Q111V can be reached by single substitutions via intermediates Q111L and Q111E, both of which are observed among multiple CG-adapted insects (*Figure 1A*). In contrast, Q111T, which has arisen independently at least three times across insect orders, require passing through states Q111P or Q111K via single nucleotide substitutions, but neither are observed among the taxa surveyed (*Supplementary file 1*). The lack of observed instances of Q111P or Q111K suggests that these states may be particularly deleterious and are either evolutionarily short-lived or circumvented altogether by dinucleotide mutation events. The latter scenario is made more plausible by recent work in *Drosophila* indicating that mutinucleotide mutation events are more frequent than previously appreciated (*Assaf et al., 2017*).

To the extent that *D. melanogaster* represents a typical CG-sensitive species, it is clear that substitutions at positions 111 and 122 would not be permitted to fix in the species without being preceded by A119S or an equivalent permissive substitution. A119S was not detected in extensive mutagenesis screens (*Zhen et al., 2012*) and has only modest effects on CG-inhibition of NKA in vitro (*Figure 2*). Given these findings and its position and orientation in the NKA-ouabain co-crystal structure (*Figure 1—figure supplement 4*), one might assume that A119S is a neutral (or nearly-neutral) permissive substitution that renders some species candidates for the adaptation to CGs by chance. However, our ability to examine the effects of A119S, and interacting substitutions, in the context of whole animal phenotypes reveals a remarkably different picture. In particular, we show that homozygosity for A119S results in substantial levels of neural dysfunction (*Figure 3*). Despite these detrimental effects, we also find that A119S, even when heterozygous, confers a substantial survival advantage upon exposure to CGs. Such insights could only be provided by evaluating the effects of mutations in the context of the whole organism. Investigating just one aspect of phenotype, for example biochemistry in vitro, yielded a misleading picture of the fitness advantages, and disadvantages, that likely operate on the individual amino acid substitutions underlying this adaptation.

Our findings raise the question of how, given its deleterious effects, A119S became established in so many species. Initially, a survival advantage conferred by A119S in CG-rich environments may have outweighed deleterious effects associated with homozygosity for this substitution. This may be especially true for insects that specialized on CG-containing plants, where competition with other species for resources is reduced. In multiple lineages, the substitutions A119S and A119N preceded substitutions to 111 and 122 (Hemiptera, Hymenoptera, Diptera). In Drosophila, where we have the greatest phylogenetic resolution, A119S was established before substitutions to sites 111 and 122 in the evolutionary lineage leading to *D. subobscura*, which appears to be polymorphic with respect to CG-insensitivity. This said, Q111L is sometimes observed in the absence of A119S (as is Q111E, see *Figure 1A*: *Bemisa tabaci* and *Lophocarya caryae*; not shown are *Aedes aegypti*, *Mechanitis polymia* and *Empyreuma pugione*, see *Supplementary file 1*) implying alternative permissive substitutions in these taxa. More generally, A119S is a common substitution among taxa that do not specialize on CG-containing hostplants (*Supplementary file 1*), again implying the existence of additional compensatory substitutions in these species. None of the remaining 20 substitutions distinguishing the *D. subobscura* and *D. melanogaster* proteins appear among sites most highly correlated with A119S in a phylogenetic analysis, however, leaving us few clues about promising candidates (*Figure 1—figure supplement 3*).

This said, a model invoking serial substitution of A119S and compensatory mutations is not the only plausible scenario. Alternatively, since CG-insensitive haplotypes substitutions including A119S lack strong deleterious effects and may confer a survival advantage upon exposure to CGs when heterozygous (*Figure 3*, *Figure 4*, *Figure 4—figure supplement 1*), such haplotypes may be maintained over time by heterozygote advantage and subsequently reach fixation as they accumulate a sufficient number of compensatory substitutions (*Kimura, 1985*) or by gene duplication (*Spofford, 1969*), which has occurred repeatedly in CG-specialist species (*Zhen et al., 2012*; *Petschenka et al., 2017*; *Yang et al., 2019*). Interestingly, *D. subobscura* is polymorphic for substitutions at positions 111 and 122 mediating CG-insensitivity and these haplotypes are associated with polymorphic inversions that may rarely be homozygous (*Pegueroles et al., 2016*). The evolution

of CG-insensitivity in *D. subobscura* may be recent and represent a transient stage in the evolution of this adaptive trait. Another alternative is that protein dysfunction associated with CG-insensitivity substitutions is in some cases compensated by the up-regulation of ATPα1 expression (see for e.g. *Loehlin et al., 2019*), which may be an important driver of recurrent ATPα1 duplication and is observed across multiple CG-adapted insects (*Yang et al., 2019*).

Our study adds to a growing body of work linking epistasis to the dynamics of adaptive evolution in contexts where the ecological relevance of the phenotype is well-understood (*Tarvin et al., 2017*; *McGlothlin et al., 2016*; *Tufts et al., 2015*; *Kumar et al., 2017*). To date, however, epistasis among substitutions in a protein has been evaluated in silico, in vitro, and in microbes along the dimensions of either protein folding stability, ligand specificity, enzyme activity, or cell growth rates (*Storz, 2016*). While it is tempting to use one axis of the mutation-phenotype map as a convenient proxy for fitness (e.g. in vitro studies of the effects of mutations on protein function), the actual effect of a given mutation on fitness is a convolution over multiple phenotypes, from the levels of enzyme and tissue function to that of whole organism fitness in its environment. By considering phenotypic effects both in vitro and in vivo, we demonstrate how dominance and epistasis of amino acid substitutions can be missed or mischaracterized in the absence of a whole animal model to evaluate these effects. As such, our study reveals surprising new features of adaptive amino acid substitutions that were previously inaccessible.

# Materials and methods

### Key resources table

| Reagent type (species) or resource | Designation | Source or reference | Identifiers | Additional information |
|---|---|---|---|---|
| Gene (*Drosophila melanogaster*) | ATPalpha1 | NA | FLYB:FBgn0027548 | |
| Strain, strain background (*D. melanogaster*) | ATPalpha1 △2-6b founder line, w[1118];;ATPa△2-6b attP/TM6B,Tb[1] | this paper | N/A | available on request |
| Strain, strain background (*D. melanogaster*) | 15 engineered variants of ATPalpha1 | this paper | N/A | See *Supplementary file 2*, available on request |
| Genetic reagent (*D. melanogaster*) | y[1], w[67c23], P{y[+mDint2]=Crey}1b;;D[*]/TM3, Sb[1] | Bloomington Drosophila Stock Center | BDSC:851 FLYB:FBst0000851; RRID:BDSC_851 | Source of Cre protein |
| Genetic reagent (*D. melanogaster*) | w[*];;ry[506] Dr[1]/TM6B, P{w[+mC]=Dfd-EYFP}3, Sb[1],Tb[1],ca[1] | Bloomington Drosophila Stock Center | BDSC:8704 FLYB:FBst0008704; RRID:BDSC_8704 | Balancer, chr 3 |
| Sequence-based reagent | PCR/sequencing primers, Sanger-based sequencing | this paper | N/A | ATPa1 H1-H2 Forward: AAAACTTGGAGCGCGATGGT; ATPa1 H1-H2 Reverse: ATACGGTCGCCGAACTTCAC |
| Sequence-based reagent | PCR/sequencing primers for Illumina-based sequencing | this paper | N/A | ATPa1 H1-H2 Forward 2834: GTCTCGTGGGCTCGG CGCTTTCAGACACATCCCGA; ATPa1 H1-H2 Forward 3185: TCGTCGGCAGCGTCTGAG AAAATGCCCGTCACGA |
| Sequence-based reagent | PCR-added linker primers, Illumina-based sequencing | this paper | N/A | ATPa1 H1-H2 i5: AATGATACGGCGACCACC GAGATCTACACnnnnnnnn TCGTCGGCAGCGTCA; ATPa1 H1-H2 i7: CAAGCAGAAGACGGCATACGAGAT nnnnnnnnGTCTCGTGGGCTCGG; where 'nnnnnnnn'=any sequence index |

*Continued on next page*

*Continued*

| Reagent type (species) or resource | Designation | Source or reference | Identifiers | Additional information |
|---|---|---|---|---|
| Chemical compound, drug | Ouabain | Sigma | Cat#: O3125 | |
| Chemical compound, drug | Porcine sodium potassium ATPase, cerebral cortex | Millipore-Sigma | Cat#: A7510 | |
| Recombinant DNA reagent | *pGX-attB-ATPa2-6b* | this paper | N/A | available on request |
| Software, algorithm | PAML v4.8 | PMID: 17483113 | RRID:SCR_014932 | http://abacus.gene.ucl.ac.uk/software/paml.html |
| Software, algorithm | BayesTraits v3.01 | PMID: 16685633 | RRID:SCR_014487 | http://www.evolution.rdg.ac.uk/BayesTraitsV3.0.1/BayesTraitsV3.0.1.html |
| Other | multiple sequence alignment (protein) | this paper | N/A | *Supplementary file 3* |
| Other | multiple sequence alignment (nucleotide) | this paper | N/A | *Supplementary file 4* |

## Sequencing and alignments

Sources of data used in this study are detailed in *Supplementary file 2*. We collected new data for three species: *Drosophila subobscura* (Diptera), *Monophadnus latus* (Hymentoptera), and *Syntomeida epilais* (Lepidoptera). In each case, total RNA was extracted using TRIzol (Ambion, Life Technologies) following the manufacturer's protocol. RNA-seq libraries were prepared with either the TruSeq RNA Library Prep Kit v.2 (Illumina) or TruSeq Stranded RNA Library Prep Gold (Illumina) and sequenced on HiSeq2500 (Genomics Core Facility, Princeton, NJ, USA). 50–55 million paired-end 150 nucleotide reads per library were trimmed for quality (Phred quality $\geq$20) and length ($\geq$30 contiguous bases). Trinity 2.2.0 (*Haas et al., 2013*) was used for de novo transcriptome assembly using default parameters. The ATPα1 coding sequences of *D. melanogaster* (FBgn0002921) and *B. mori* (GenBank: LC029030.1) were used to query the assembled transcripts using either tblastx or tblastn (blast-2.26). For *Dysdercus cingulatus* (Hemiptera), *Macrocheraia grandis* (Hemiptera), *Largus californicus* (Hemiptera), *Jalysus sp.* (Hemiptera), *Metatropis rufescens*, *Ischnodemus falicus*, *Geocoris sp.*, *Gyndes sp.*, *Monomorium pharaonis*, *Hylobius abietis*, *Megacyllene caryae*, *Calligrapha philadelphica*, and *Basilepta melanopus*, we downloaded raw RNA-seq data, carried out de novo assembly and obtained ATPα1 sequences using methods similar to those described above. The remaining ATPα1 sequences were either previously published or obtained from publicly available genome references assemblies. A multiple nucleotide sequence alignment was created using ClustalOmega (*Sievers et al., 2011*). Two alternatively spliced exons were masked for subsequent phylogenetic analyses. Alignments are available as *Supplementary file 2*.

## Statistical phylogenetic analyses

Phylogenetic relationships were established based on previously published sources (*Supplementary file 2*). Phylogeny branch lengths were estimated using IQtree (*Nguyen et al., 2015*) on the nucleotide alignment of 174 ATPα1 sequences representing 161 species sampled from eight insect orders (*Supplementary file 1*) using a guide phylogeny (*Figure 1—figure supplement 2*) to force species branching order within orders. We carried out a comprehensive search for substitutions at ATPα1 on which substitutions at sites 111 and 122 were dependent using BayesTraits (version 3.01; *Pagel and Meade, 2006*). We first reconstructed the ancestral protein sequences of ATPα1 using the codeml(aaml) algorithm of PAML (version 4.8a; *Yang, 2007*). As inputs, we provided PAML with the phylogeny with branch lengths and the protein sequence alignment for ATPα1. We used the default PAML parameters with the following changes: 'cleandata' was set to 0 to preserve sites with missing data; 'fix_blength' was set to one to use the branch lengths as the initial values in the ancestral reconstruction; 'method' was set to one to use a new algorithm in PAML that updates branch lengths one-by-one. Using the inferred ancestor, we then binarized each amino acid

state in the multi-species alignment into ancestral, '0', and derived, '1', and used this, and the phylogeny with branch lengths as the input for BayesTraits. BayesTraits fits a continuous-time Markov model to estimate transition rates between discrete, binary traits and calculates the likelihood of the fitted model. Restricting rate parameters appropriately (see below), we tested whether the transition rates for sites 111 and 122 were dependent on the state at all other variant sites. We excluded 56 sites with just a single substitution (i.e. one instance in 174 sequences in the alignment) due to their low information content.

In all models, double transition rates (e.g. Q111V and A119S occurring at the same time) were set to zero as double transitions are highly unlikely to occur in a single step and can be modeled as two single transitions, following *Pagel (1994)*. Additionally, we set all transition rates from the derived state back to the ancestral state to zero as failing to do this resulted in unrealistically high estimates of the reversion rate. After these restrictions were imposed, the null model (i.e. independence) had four transition parameters. To test the dependence between sites, we refit an alternative model with two additional restrictions: one forcing the transition rate at site one to be fixed regardless of the state of site 2, and a second forcing the transition rate at site two to be fixed regardless of the state of site 1. This effectively tests whether the transition rate is affected by the state of either site. Following the BayesTraits manual recommendations, the phylogeny branch lengths were scaled using BayesTraits to have a mean length of 0.1. Additionally, to increase the chance of finding the true maximum likelihood, we set MLTries to 250 which controls the number of times BayesTraits calls the maximum likelihood algorithm, returning the maximum likelihood of the 250 attempts. We ran each substitution pair 25 times and checked that at least half of the runs had equivalent maximum likelihoods (within 0.1) to ensure that the results were stable. Taking the maximum likelihood for each pair, for each model, we calculated p-values using the likelihood ratio test (LRT), where the statistic of merit is: 2 (unrestricted model – restricted model). The LRT statistic is chi-squared distributed with degrees of freedom (df) equal to the number of restrictions imposed on the model (in this case, df = 2).

A potential concern is that the models employed by PAML codeml(aaml) and BayesTraits ignore hierarchical relationships among codons encoding amino acid substitutions. This is a concern because, under a single nucleotide substitution model, Q111V must be reached by passing through states Q111L or Q111E. Likewise, Q111T must be reached by passing through states Q111P or Q111K. Despite ignoring these relationships, the effect on our analysis is likely to be minimal. First, although it is clear from phylogenetic patterns that Q111V was reached via Q111L in Danaid butterflies and Chrysomelid beetles (*Aardema et al., 2012*), there is no clear phylogenetic evidence indicating that Q111V or Q111T have passed though intermediate states in other instances. This implies that intermediate states may be short-lived and that modeling these events as single steps may be a reasonable approximation. Second, the occurrence of multi-nucleotide mutations (e.g. generating Q111V or Q111T via a single mutation event) may be more frequent than commonly assumed (*Assaf et al., 2017*), and even PAML's codon-based model (codonml) does not account for this. Third, ignoring hierarchical relationships among amino acid substitutions is more likely to lead to spurious conclusions about substitution order (which we do not consider) than co-occurrence. Finally, we use BayesTraits as one of several tools to identify key substitutions and guide functional experiments. Our focus on specific correlations does not imply that other correlations are not important, nor that some important functional interactions will be missed.

## Engineering ATPα amino-acid substitution lines

To test for phenotypic effects associated with candidate amino-acid substitutions, we developed a genetic engineering approach of the endogenous *ATPα1* (CG5670) locus in *Drosophila melanogaster*. Employing a strategy similar to that of *Roland et al. (2013)*, we first generated a 'founder line' by deletion of exons 2-6b and replacement with a functional *attP* element using ends-out homologous recombination (*Figure 1—figure supplement 1*). To generate the founder line, homology arms were generated as 2–3 kilobase PCR amplicons 5' of exon 2 and 3' of exon 6b and inserted into the *pGX-attP* vector (*Huang et al., 2009*). This construct was injected into *w1118* embryos using standard protocols for P-element transgenesis (Rainbow Transgenic Flies, Inc). Ends out recombination was performed and molecularly confirmed. The *mini-white* gene was subsequently removed by Cre-lox mediated excision (*Groth et al., 2004*) to create the founder line, *w1118;;ATPαΔ2-6b attP/ TM6B,Tb1* (exons 2-6b were replaced by a functional *attP* site).

To generate allelic variant lines of ATPα1, we cloned exons 2-6b into *pGX-attB-ATPα2-6b* vector and injected it into the *attP* founder line mated to *vasa-phiC31-ZH2A* flies using standard protocols (Rainbow Transgenic Flies, Inc). The unmodified plasmid was used to generate a wild-type control (*GE-ATPα−Δ2-6b-WT*) following Cre-lox reduction of the *mini-white* using Bloomington Drosophila Stock Center #851 (y[1], w[67c23], P{y[+mDint2]=Crey}1b;; D*/TM3, Sb[1]) as a source of Cre. We then used site-directed mutagenesis (Quick-change Lightening, Agilent) to generate a panel of amino-acid substitution variants of *pGX-attB-ATPα2-6b* and corresponding GE strains were generated and Cre-lox reduced, as above for the WT control (*Supplementary file 2*). Variant and control lines were balanced by crossing to w*;;ry[506] Dr[1]/TM6B, P{w[+mC]=Dfd-EYFP}3, Sb[1],Tb[1],ca[1] (Bloomington Drosophila Stock Center #8704) and selecting for EYFP florescence. Homozygous variant GE-ATPα lines were established, when possible, by sib-mating and selection of non-florescent Tb[+],Sb[+] larvae. All transgenic strains were generated using the same method and all substitutions were confirmed using PCR and Sanger-based sequencing.

## Viability assays

Viability was measured in three ways. First, we measured viability as the relative hatchability of embryos carrying a given homozygous amino acid substitution (hereafter 'Mut'). To do this, we self-crossed each balanced *GE-ATPα−Δ2-6b-Mut/TM6B*, P{w[+mC]=Dfd-EYFP}3, Sb[1],Tb[1],ca[1] and allowed them to lay fertilized eggs on standard apple juice–agar Petri plates with yeast paste for two hours at 25°C, 50% humidity. After 24 hr, the relative proportion of non-EYFP (i.e. homozygous *Mut/Mut*) to EYFP (i.e. heterozygous *Mut/+*) first instar larvae were counted. As a control, we generated *GE-ATPα−Δ2-6b-WT/TM6B*, P{w[+mC]=Dfd-EYFP}3, Sb[1],Tb[1] parents and followed the same procedure. A second measure of viability is the probability of survival of a first instar *Mut/Mut* larvae to adulthood. To measure this, we transferred homozygous (*Mut/Mut*) first instar larvae to fly media vials (recipe R, LabExpress, Ann Arbor, MI). Larval density was limited to 10 per vial and vials were kept at 25°C, 50% humidity. After two weeks, we counted the number of emerging adults. Homozygous *GE-ATPα−Δ2-6b-WT* first instar larvae generated the same way were used as a control. Third, we measured egg-to-adult fitness as the proportion of emerging *Mut/Mut* adults in media bottles (recipe B, LabExpress, Ann Arbor, MI) seeded with *GE-ATPα−Δ2-6b-Mut/TM6B*, P{w[+mC]=Dfd-EYFP}3, Sb[1],Tb[1],ca[1] parents and used bottles seeded with +/TM6B, P{w[+mC]=Dfd-EYFP}3, Sb[1], Tb[1],ca[1] parents as controls.

## Enzyme inhibition assays

For each *D. melanogaster* substitution line, we homogenized 90 heads (previously stored at −80°C) in 900 µl of deionized water, using a 1 ml glass grinder (Wheaton) chilled on ice (*Petschenka et al., 2017*). After vortexing, we divided homogenates into three aliquots of 300 µl representing technical replicates. Subsequently, samples were frozen at −80°C and freeze-dried (Christ, Alpha 2–4 LDPlus) overnight. For assessing resistance of NKA to ouabain, we followed the procedure as described in *Petschenka et al. (2013a)* that is based on the photometric evaluation of phosphate released from enzymatic hydrolysis of adenosine triphosphate (ATP) as a measure of NKA activity. Lyophilisates were stored at −80°C until use and reconstituted with 500 µl deionized water. Head extractions were incubated at 12 increasing concentrations of the water-soluble standard cardenolide ouabain (range $10^{-8}$ M to $2 \times 10^{-3}$ M, Sigma, Germany, O3125) at 37°C under the following conditions: 100 mM NaCl, 20 mM KCl, 4 mM MgCl$_2$, 50 mM imidazol, and 2.5 mM ATP (pH 7.4). We corrected all measurements for a background value that we obtained by incubating the extract under the same conditions as above, except with the addition of $2 \times 10^{-3}$ M ouabain and no KCl (i.e., NKA inactive). On each microplate, we included an assay with porcine Na[+]/K[+]-ATPase (Sigma, Germany, A7510-5UN) as an internal standard. In addition, we ran a series of KH$_2$PO$_4$ dilutions as a phosphate calibration curve on every plate. We measured absorbances at 700 nm using a CLARIOstar microplate reader (BMG Labtech, Germany). We carried out three biological replicates per line based on different extractions of fly heads. Each biological replicate was the average of three technical replicates, that is measurements based on aliquots from original extracts. For data analysis, we compared all measurements to a non-inhibited control.

To quantify levels of insensitivity to CG-inhibition, we used least-squares fitting to the bi-phasic curve function,

$$A = (1-f)\left(1 - \frac{[I]}{[I]+IC_{50,1}}\right) + f\left(1 - \frac{[I]}{[I]+IC_{50,2}}\right), \tag{1}$$

where $IC_{50,1}$ and $IC_{50,2}$ represent the inhibitor concentration [I] required for 50% activity $A$ of each isoform present in proportions (1 $f$) and $f$, respectively. For heterozygous lines, we expect a biphasic curve reflecting the inhibitor dissociation constants ($K_d$) of the two forms of the enzyme (*Figure 2—figure supplement 1A*). If the two forms of the enzyme contribute equally to the total ouabain-inhibitable activity, we expect $f = 0.5$. For homozygous lines, we estimated a single $IC_{50}$ by setting $f = 0$ in the above equation, which assumes a homogenous population of enzyme. Least-squares fitting was performed with the nlsLM function of the minpack.lm library in R. Alternatively, we used Origin-Pro 2017 (OriginLab, Northampton, MA) with top and bottom asymptotes set to 100 and 0, respectively. The two approaches yielded similar results. Confidence intervals on estimates of $IC_{50,1}$, $IC_{50,2}$ and $f$ were estimated using parametric bootstrap simulations, assuming that residuals across biological replicates are normally distributed. We assumed mixtures exhibited monophasic curves if the confidence intervals for $f$ included 0.

As validation of the approach, we compared inhibition curves for the commercial pig NKA enzyme (porcine cerebral cortex, Sigma A7510), extracts from Monarch butterfly brains, and normalized mixtures of the two (*Figure 2—figure supplement 1*, panels B and C). Enzyme-inhibition activities were analyzed as above except using six ouabain concentrations (range $10^{-8}$ M to $10^{-3}$ M) and one biological replicate. Parameters for homogenous extracts and mixtures were estimated using the monophasic ($f = 0$) and biphasic models ($f > 0$), respectively. As expected, the pig and Monarch enzymes exhibit monophasic curves whereas the mixtures exhibit biphasic curves with $f$ estimates close to those expected based on mixture proportions.

## Targeted allele-specific expression

To estimate relative expression of mutant and wild-type alleles in heterozygous strains, we designed a targeted allele-specific expression (ASE) assay. Genomic DNA (gDNA) and total RNA were extracted sequentially using TRIzole Reagent (Invitrogen 15596026 and 15596018) from three replicate pools of 20 males sampled from each of sixteen lines. Extracted RNA was further purified with Qiagen RNAeasy column (Qiagen 74104) and reverse transcribed into cDNA with the use of random primers and ProtoScript II Reverse Transcriptase (NEB M0368S). Two sets of primers for ATPα1 were designed and used in subsequent PCR, multiplexing and sequencing. The first set were designed as exon-primed intron crossing primers (*Racle et al., 2017*) and span a short intron separating exons 2 and 3. A second set of primers was used to add standard Illumina-like i5 and i7 barcodes to the PCR amplicons to facilitate multiplexing and subsequent sequencing. These multiplexed amplicons were pooled and sequenced on an Illumina MiSeq (Princeton Microarray Facility) and yielded ~1 million 150 nucleotide paired-end reads. Reads were mapped to gDNA and cDNA reference sequences using bwa-mem (*Li, 2013*) and a Variant Calling File (VCF) file was produced using the Naive Variant Caller as implemented in Galaxy (Version 0.0.2). ASE was estimated from allele counts at focal sites (104, 111 and 122) using the Cochran-Mantel-Haenzel framework as implemented in R (mantelhaen. test). Specifically, ASE is estimated as the relative risk of the mutant substitution in the cDNA population using allele counts from gDNA as a reference population.

## Behavioral phenotyping

We quantified 'bang sensitivity' which is a measure of susceptibility to seizures and paralysis upon mechanical over-stimulation (*Ganetzky and Wu, 1982*). As such, the bang sensitivity phenotype is a measure of proper neuron function. Individual flies were placed in an empty fly media vial and vortexed at the maximum setting for 20 s. Immediately following mechanical overstimulation, neurologically dysfunctional flies typically experience a period of convulsions and seizures. The recovery time was recorded as the time for each fly to right itself. Male flies were assayed 14 days post-eclosion. Approximate 95% confidence intervals were estimated around means by bootstrap resampling with replacement. An average of 43 flies were assayed (range 37–85). Recovery time distributions were compared using a Wilcoxon two-sample rank sum test with continuity correction as implemented in R (wilcox.test).

## CG exposure assay

As a measure of the ability to tolerate CGs, we exposed adult flies (1–7 days post-eclosion) to media containing known concentrations of the CG ouabain (Sigma). 1.5 grams of *Drosophila* instant media (Carolina Biological Supply) was reconstituted in a plastic vial with 7 mL of either 0, 5, 10 or 20 mM ouabain. After food solidified (30 min), a small paper wick was added. Three replicates, each with 10 males and 10 females, were kept in separate vials at 25°C and 50% humidity. Mortality was measured after 7 days. The genetically engineered wild-type line was used as a control. The relationship between treatment concentrations and physiologically-relevant concentrations for animals feeding on CG-containing plants is difficult to establish. Compared to values reported for various milkweed species (*Züst et al., 2019*), our 10 mM CG treatment corresponds to a per mg dry weight concentration that is ~10 fold higher than the range reported for leaf tissue, but well within those reported for milkweed latex. The Cochran-Mantel-Haenzel test framework (implemented in R) was used to assess significant differences between treatments (i.e. 5/10/20 mM ouabain versus no ouabain), as well as estimates and 95% confidence bounds for the relative risk associated with treatment. Odds Ratios (*OR*) estimated using this framework were converted into relative risk (*RR*) estimates using the formula, $RR = OR/(1 - p + (p * OR))$, where $p$ is the risk in the no ouabain control group. A constant (0.5) was added to all cells to allow for calculation of relative risk in cases where mortality or survivors were absent. Thus, the maximum relative risk for this sample size (three replicates of 20 individuals each) is limited to 117. We found no sex differences in survival.

## Acknowledgements

We thank M Przeworski and M Schumer for comments on the manuscript, D James and A Wilson for sending *Syntomeida epilais* specimens and M Aguadé for providing pre-publication access to the *D guanche* ATPα1 sequence. We thank Sabrina Stiehler and Miyoung Jang for technical assistance. This work was funded by NIH R01 GM115523 to PA, NIH T32 GM008424 to BPR, NIH R01 GM108073 and NIH R01 AG027453 to MJP, and DFG PE 2059/3-1 and the LOEWE program of the State of Hesse (Insect Biotechnology & Bioresources), Germany to GP.

## Additional information

### Funding

| Funder | Grant reference number | Author |
| --- | --- | --- |
| National Institutes of Health | R01 GM115523 | Peter Andolfatto |
| National Institutes of Health | T32 GM008424 | Bartholomew P Roland |
| National Institutes of Health | R01 GM108073 | MIchael J Palladino |
| National Institutes of Health | R01 AG027453 | Michael J Palladino |
| Deutsche Forschungsgemeinschaft | PE 2059/3-1 | Georg Petschenka |
| LOEWE program of the State of Hesse | Insect Biotechnology & Bioresources | Georg Petschenka |

The funders had no role in study design, data collection and interpretation, or the decision to submit the work for publication.

### Author contributions

Andrew M Taverner, Resources, Data curation, Investigation, Methodology, Writing—review and editing; Lu Yang, Resources, Data curation, Formal analysis, Investigation, Methodology, Writing—review and editing; Zachary J Barile, Becky Lin, Arya S Rao, Daniel Wei, Investigation; Julie Peng, Bartholomew P Roland, Aaron D Talsma, Investigation, Writing—review and editing; Ana P Pinharanda, Supervision, Investigation, Writing—review and editing; Georg Petschenka, Formal analysis, Investigation, Methodology, Writing—review and editing; Michael J Palladino, Resources, Supervision, Funding acquisition, Validation, Investigation, Methodology, Writing—review and

editing; Peter Andolfatto, Conceptualization, Resources, Software, Formal analysis, Supervision, Funding acquisition, Validation, Investigation, Visualization, Methodology, Writing—original draft, Project administration, Writing—review and editing

### Author ORCIDs
Andrew M Taverner https://orcid.org/0000-0001-8265-6836
Lu Yang https://orcid.org/0000-0002-2694-1189
Ana P Pinharanda https://orcid.org/0000-0002-4696-9218
Arya S Rao http://orcid.org/0000-0003-3007-4812
Peter Andolfatto https://orcid.org/0000-0003-3393-4574

### Decision letter and Author response
Decision letter https://doi.org/10.7554/eLife.48224.028
Author response https://doi.org/10.7554/eLife.48224.029

## Additional files

### Supplementary files
• Supplementary file 1. Summary of amino acid variation at sites implicated in CG-sensitivity of ATPαone in surveyed species. Columns correspond to 35 sites implicated in cardenolide-sensitivity with the addition of six additional sites of interest (112, 114, 119, 787, 874, 898). Following convention, position is standardized relative to the sheep (*Ovis aries*) sequence NM_001009360 minus five amino acid residues from 5'end. The reference sequence refers to the consensus sequence among non-specialist species. A dot indicates identity with the reference sequence and dashes indicate missing data. Species highlighted in green are specialists on cardiac glycoside (CG) containing plants.
DOI: https://doi.org/10.7554/eLife.48224.004

• Supplementary file 2. Includes sources of sequence data used in this study, references for phylogenetic relationships a list of transgenic strains generated in this study.
DOI: https://doi.org/10.7554/eLife.48224.017

• Supplementary file 3. Fasta nucleotide sequence alignment of all sequences used in this study.
DOI: https://doi.org/10.7554/eLife.48224.018

• Supplementary file 4. Fasta amino acid sequence alignment of all sequences used in this study.
DOI: https://doi.org/10.7554/eLife.48224.019

• Transparent reporting form
DOI: https://doi.org/10.7554/eLife.48224.020

### Data availability
Sequence data as been deposited in Genbank and the details of all accession numbers of this and previously published data are tabulated in Supplementary file 2.

The following datasets were generated:

| Author(s) | Year | Dataset title | Dataset URL | Database and Identifier |
|---|---|---|---|---|
| Taverner T, Yang L, Barile ZJ, Lin B, Peng J, Pinharanda A, Rao A, Roland BP, Talsma AD, Wei D, Petschenka G, Palladino MJ, Andolfatto P | 2019 | RNA-seq data for *Monophadnus latus* (Hymenoptera) | https://www.ncbi.nlm.nih.gov/sra/?term=SRR9586628 | NCBI SRA, SRR9586628 |
| Taverner T, Yang L, Barile ZJ, Lin B, Peng J, Pinharanda A, Rao A, Roland BP, Talsma AD, Wei D, Petschenka | 2019 | RNA-seq data for *Drosophila subobscura* (Haplotype: QSN) | https://www.ncbi.nlm.nih.gov/sra/?term=SRR9586630 | NCBI SRA, SRR9586630 |

G, Palladino MJ,
Andolfatto P

| Taverner T, Yang L, Barile ZJ, Lin B, Peng J, Pinharanda A, Rao A, Roland BP, Talsma AD, Wei D, Petschenka G, Palladino MJ, Andolfatto P | 2019 | RNA-seq data for *Syntomeida epilais* (Lepidoptera) | https://www.ncbi.nlm. nih.gov/sra/?term= SRR9586629 | NCBI SRA, SRR9586629 |
|---|---|---|---|---|

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
