## [Decision Letter]

Thank you for submitting your article "Adaptive substitutions underlying cardiac glycoside insensitivity in insects exhibit epistasis in vivo" for consideration by *eLife*. Your article has been reviewed by three peer reviewers, including Lauren A O’Connell a the Reviewing Editor and Reviewer #1, and the evaluation has been overseen by Patricia Wittkopp as the Senior Editor. The following individuals involved in review of your submission have agreed to reveal their identity: Harold H Zakon (Reviewer #2); Virginie Courtier-Orgogozo (Reviewer #3).

The reviewers have discussed the reviews with one another and the Reviewing Editor has drafted this decision to help you prepare a revised submission.

Summary:

This is an interesting paper about the evolution of resistance in insects to the toxic cardiac glycosides utilized by some plants as a herbivore defense. This is a rich and well ploughed field of research, but this paper, because it approaches the topic in a deeper and more integrative fashion than usual, makes novel and important observations. In this field, typically a gene for the target - Na+/K^+^ ATPase (NKA) - is sequenced, amino acid (AA) substitutions in the protein are inferred, and their ability to confer toxin resistance is tested in vitro by expression and site-directed mutations. This study by Taverner et al. starts out simply enough by testing whether the two most frequent and intensively studied substitutions in NKA that protect against toxin binding are neutral as far as function of NKA in the whole animal. They engineered these AA substitutions into the single NKA of flies and found, surprisingly, that these were recessive lethal. They then undertook a number of logical experiments to understand how these AA substitutions might be selected if they are so detrimental when expressed in the whole animal. They found a previously overlooked substitution at a site that contributes little to toxin resistance but "corrects" the deleterious substitutions, although not completely so that even flies that are heterozygous for this substitution are impacted. They then asked if the AA substitutions that are typically studied in vitro act in vivo. They fed the engineered flies on various concentrations of ouabain in their food and observed that, indeed, the substitutions saved flies from the deleterious effects of this NKA inhibitor. Finally, based on all their observations, the authors pose a model in which NKA with some AA substitutions occurs in heterozygosity in those insects that feed on cardioglycoside-rich plants where, despite the sub-optimal NKA, the advantages of toxin protection outweigh the detriment of suboptimal NKA enzyme activity. Eventually, as a result of further AA substitutions on this enzyme or, as often happens, gene duplication leaving an optimally-functioning but cardioglycoside-unprotected paralog working in the brain and a cardio-glycoside protected analog in the gut, a more evolutionarily stable situation is reached.

Essential revisions:

The reviewers raise a number of concerns that must be adequately addressed before the paper can be accepted. These required revisions only require clarifications in the text and do not require further experimentation.

1) At position 111 and 112 the authors investigated three amino acid changes, Q111V, Q111T and N122H. Whereas N122H can arise via a single nucleotide substitution, the other two amino acid changes require at least two nucleotide substitutions according to the genetic code. Can the authors elaborate a bit on this point? Is this fact taken into account in their statistical phylogenetic analysis using PAML and BayesTraits? If not, could it affect the results of the statistical phylogenetic analysis? Also, did the amino acid changes Q111V and Q111T occur via two successive nucleotide substitutions (i.e. are there intermediate DNA sequences found in natural populations or intermediate species)? If so, what would be the intermediate amino acid? Wouldn't it be important in the future to test the effect of the two successive amino acid changes on fitness?

2) This paper only considers AA substitutions but not levels of expression. While NKA functions sub-optimally in engineered flies it may be the case that, in nature, changes in NKA gene expression could compensate for sub-optimal enzyme efficiency. Admittedly, this would still be costly as more energy would have to go into generating higher concentrations of enzyme. The authors should address this point.

3) With the methods at the end, more information is needed for a wide readership to understand the enzyme inhibition assays and the expected biphasic inhibition curves.

---

## [Author Response]

Essential revisions:The reviewers raise a number of concerns that must be adequately addressed before the paper can be accepted. These required revisions only require clarifications in the text and do not require further experimentation.1) At position 111 and 112 the authors investigated three amino acid changes, Q111V, Q111T and N122H. Whereas N122H can arise via a single nucleotide substitution, the other two amino acid changes require at least two nucleotide substitutions according to the genetic code. Can the authors elaborate a bit on this point? Also, did the amino acid changes Q111V and Q111T occur via two successive nucleotide substitutions (i.e. are there intermediate DNA sequences found in natural populations or intermediate species)? If so, what would be the intermediate amino acid? Wouldn't it be important in the future to test the effect of the two successive amino acid changes on fitness?

This is an interesting point. We initially excluded discussion of this point to keep the Discussion more focused, but since this has been highlighted by the reviewers, we have added back in a short paragraph discussing this observation and some of its implications (see Discussion, second paragraph). Assuming single nucleotide mutations, Q111V could be reached by intermediate states Q111L or Q111E, both of which are observed in other taxa. The case of Q111T is more interesting as it requires passing through states Q111P or Q111K via single nucleotide substitutions, which are not observed in other taxa. The absence of Q111P and Q111K in other taxa suggests that these states are particularly deleterious, which would definitely be interesting to test. However, an alternative hypothesis is that Q111T could be reached in a single step by dinucleotide mutation events, which are more frequent than previously appreciated (Assaf et al., 2017).

Is this fact taken into account in their statistical phylogenetic analysis using PAML and BayesTraits? If not, could it affect the results of the statistical phylogenetic analysis?

This is valid concern and we agree that it deserves more justification. We have added the following paragraph to the Materials and methods section concerning the use of PAML and BayesTraits: “A potential concern is that the models employed by PAML codeml(aaml) and BayesTraits ignore hierarchical relationships among codons encoding amino acid substitutions. […] Our focus on specific correlations does not imply that other correlations are not important, nor that some important functional interactions will be missed.”

2) This paper only considers AA substitutions but not levels of expression. While NKA functions sub-optimally in engineered flies it may be the case that, in nature, changes in NKA gene expression could compensate for sub-optimal enzyme efficiency. Admittedly, this would still be costly as more energy would have to go into generating higher concentrations of enzyme. The authors should address this point.

We agree this is an important point and now state this explicitly in the text, and cite a recent paper by Loehlin et al., 2019, that nicely demonstrates this idea (Discussion, fifthparagraph). We also note that the pressure to up-regulate gene expression to compensate for negatively pleiotropic adaptive substitutions may be an important driver of recurrent duplication, which is observed across multiple CG-adapted taxa.

3) With the methods at the end, more information is needed for a wide readership to understand the enzyme inhibition assays and the expected biphasic inhibition curves.

We have striven to make the rationale for the approach clearer in the main text (**see** subsection “A119S rescues enzyme dysfunction associated with substitutions at sites 111 and 122”). In addition, we have re-worked the Materials and methods section describing this analysis to improve clarity. We also added new data and results validating the approach (subsection “Enzyme inhibition assays”, Figure 2—figure supplement 1 B and C).